# Application of YOLOv5 for Point Label Based Object Detection of Black Pine Trees with Vitality Losses in UAV Data

Peter Hofinger [1,*], Hans-Joachim Klemmt [1], Simon Ecke [1,2], Steffen Rogg [3] and Jan Dempewolf [1]

[1] Department of Silviculture and Mountain Forests, Bavarian State Institute of Forestry, Hans-Carl-von-Carlowitz-Platz 1, 85354 Freising, Germany; hans-joachim.klemmt@lwf.bayern.de (H.-J.K.); simon.ecke@lwf.bayern.de (S.E.); jan.dempewolf@lwf.bayern.de (J.D.)

[2] Chair of Forest Growth and Dendroecology, Faculty of Environment and Natural Resources, University of Freiburg, Tennenbacher Str. 4, 79106 Freiburg, Germany

[3] Department of Forestry, University of Applied Sciences Weihenstephan-Triesdorf, Hans-Carl-von-Carlowitz-Platz 3, 85354 Freising, Germany; steffen.rogg@hswt.de

[*] Correspondence: hofinger-peter@gmx.de

**Abstract:** Monitoring tree diseases in forests is crucial for managing pathogens, particularly as climate change and globalization lead to the emergence and spread of tree diseases. Object detection algorithms for monitoring tree diseases through remote sensing rely on bounding boxes to represent trees. However, this approach may not be the most efficient. Our study proposed a solution to this challenge by applying object detection to unmanned aerial vehicle (UAV)-based imagery, using point labels that were converted into equally sized square bounding boxes. This allowed for effective and extensive monitoring of black pine (*Pinus nigra* L.) trees with vitality-related damages. To achieve this, we used the "You Only Look Once" version 5 (YOLOv5) deep learning algorithm for object detection, alongside a 16 by 16 intersection over union (IOU) and confidence threshold grid search, and five-fold cross-validation. Our dataset used for training and evaluating the YOLOv5 models consisted of 179 images, containing a total of 2374 labeled trees. Our experiments revealed that, for achieving the best results, the constant bounding box size should cover at least the center half of the tree canopy. Moreover, we found that YOLOv5s was the optimal model architecture. Our final model achieved competitive results for detecting damaged black pines, with a 95% confidence interval of the $F_1$ score of 67–77%. These results can possibly be improved by incorporating more data, which is less effort-intensive due to the use of point labels. Additionally, there is potential for advancements in the method of converting points to bounding boxes by utilizing more sophisticated algorithms, providing an opportunity for further research. Overall, this study presents an efficient method for monitoring forest health at the single tree level, using point labels on UAV-based imagery with a deep learning object detection algorithm.

**Keywords:** point labels; stress detection; forest health; UAV; deep learning; YOLOv5

## 1. Introduction

Globalization and climate change impact forests world-wide by causing pathogen invasions, as well as disease emergence and spread [1]. Monitoring forest health under these changing conditions is important, as it provides insight into diseases and advice for pathogen management to be formulated [2,3].

Forest monitoring consists mainly of collecting data, which has many different approaches as it is a well-researched topic. Sampling the data in situ by foot has made significant progress, but it sometimes includes subjective measurements and is expensive. It is thus rather suited for limited sampling, but less for temporally or spatially extensive coverage [4–6]. An alternative to in situ sampling is remote sensing, which can differentiate diseased trees from healthy ones if applied correctly [7]. While mapping forest health is possible using satellite imagery [8], the low resolution prohibits automated labeling on

the individual tree level [9]. Using unmanned aerial vehicles (UAVs) can reduce costs as well as response times and can expand spatial coverage in comparison to field data [10–12]. The increase in studies using UAVs reflects these advantages [13]. The algorithms used on UAV-based imagery are diverse but most have in common that they require a dataset for training as they are machine learning or deep learning algorithms [14]. The datasets are categorized as classification, regression [15], object detection, or segmentation [16]. Classification assigns a class to previously manually or automatically segmented images [17], while regression assigns a continuous value to an image [18]. Object detection usually uses bounding boxes in an image [19,20] to capture the spatial extent and location of objects, as well as the class of objects [21]. Segmentation requires pixel-wise class labels as it classifies every pixel of an image [22] and is thus the most resource-intensive kind of labeling. While the kind of labels used varies across different use cases [23], bounding boxes are well-suited labels for monitoring tree health, as manually labeled datasets with bounding boxes enable state-of-the-art performance [24]. Additionally, bounding boxes are relatively simple compared to the labels required for segmentation and require no additional manual or automated tree crown delineation, unlike classification.

While bounding boxes (Figure 1a) are appropriate for monitoring tree health, point labels (Figure 1b) might cover all relevant information. The reason for using bounding boxes for tree health monitoring is mainly the compatibility with object detection algorithms that require bounding boxes. Bounding boxes contain information on location, spatial extent, and the class of individual objects. However, the spatial extent of individual trees may not be important for tree health monitoring, while the location and class of individual trees (e.g., degree of damage, tree species) are. The location and class of individual objects is contained within point labels. For example, point labels would contain all the information needed for monitoring the spatial distribution of damaged trees. Sometimes, point labels might be available from other investigations and thus represent low-cost data if they could be used for training an algorithm. This is significant because making a dataset for a remote sensing algorithm is still expensive, considering image collection and manual labeling. There have been attempts unrelated to tree health monitoring that tried replacing bounding boxes with point labels by creating new algorithms. Those, however, showed limitations, such as performing worse on larger objects or objects distant from each other. This topic is generally little explored [25–27]. Instead of using an algorithm that directly takes point labels as input, another option could be converting the point labels to bounding boxes first, which then are used to train an object detection algorithm requiring bounding boxes. This would allow for state-of-the-art object detection algorithms to use point labels. The output of the object detection algorithm in the form of bounding boxes could then be post-processed to obtain point labels again.

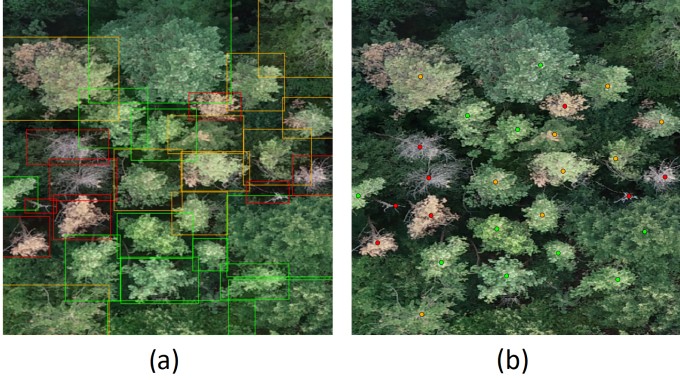

(a)　　　　　　　　　　　　　　　　(b)

**Figure 1.** Trees labeled as (**a**) bounding boxes and as (**b**) points on the same orthomosaic with healthy black pines (green), damaged black pines (orange), and dead black pines (red).

In this study, we wanted to reuse point labels of a previous investigation [28] for object detection of black pines (*Pinus nigra* L.). There, the labels were classified by the degree of vitality-related damage on high-resolution imagery over a large area using "You Only Look Once" version 5 (YOLOv5). Black pines were chosen as the object of investigation because they were considered relevant to the regionally changing climate in Germany. However, in the largest black pine forest in Germany, many trees showed signs of vitality loss. The orthomosaics were UAV-based red, green, blue (RGB) images, as high-resolution and a low cost were preferred. While using more information-rich sensors, such as multispectral sensors, might have allowed for more differentiated distinctions besides the objectives of our study, RGB cameras often perform well for monitoring tree diseases and are more cost-effective [14]. The study was conducted on the largest black pine forest in Germany, which primarily consists of a pure stand of black pine trees. According to samples previously collected, approximately 12% of the black pines of this forest were dead and 59% were damaged at the time of UAV image collection. Thus, the selected study area was suited for both training and testing the proposed method. The main goals of our study were as follows:

- To develop a new method for assessing the vitality-related damages of black pines using point labels on high-resolution UAV-based RGB imagery with YOLOv5, reducing the labeling effort;
- To identify the optimal bounding box size and model size for the proposed method, enabling an efficient conversion of point labels to bounding boxes for object detection;
- To demonstrate the competitive performance of the proposed method by comparing it to similar studies, showing its potential for practical applications.

## 2. Materials and Methods

### 2.1. Data Collection and Annotation

The study area was located in the largest black pine forest in Germany. The forest located in the municipalities of Leinach and Erlabrunn extends over 230 hectares, of which the study area covers approximately 90 hectares (centered at 49.856°N, 9.833°E). The study area included only areas with a mostly pure black pine stand. The images were collected in 2019 on 13, 15, and 16 August with a DJI Phantom 4 RTK at an altitude of 100 m with an RGB camera [28]. Those images were converted into 27 orthomosaics using photogrammetry with a spatial resolution ranging from 2 to 4 cm.

The trees were labeled in the arc geographic information system (ArcGIS) Pro within circular virtual sample plots with an area of 500 square meters each, which were systematically placed on the orthomosaics (Figure 2). We labeled every visible tree above 2 m within a sample circle by marking its highest point on a normalized digital surface model with a point label, along with its class. We used five different classes: healthy black pines, damaged black pines, dead black pines, other conifers, and deciduous trees (Figure 1b). The degree of damage was assessed by a verbal description: healthy black pines have exclusively green needles, damaged black pines have at least partially brown needles or branches that are visible due to sparse needles, and dead black pines are, at a minimum, 80% damaged.

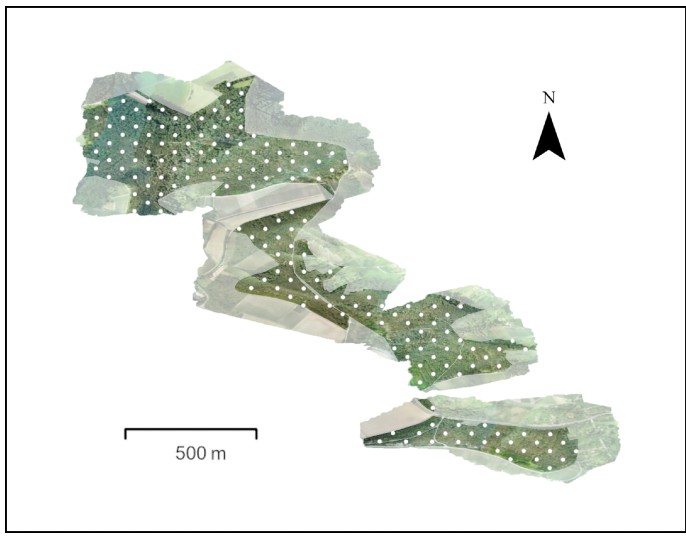

**Figure 2.** Virtual sample plots (white) on all 27 UAV-based orthomosaics of the study area (saturated).

All orthomosaics were resampled using cubic interpolation to achieve a consistent pixel size of approximately 2.8 cm in width and height. Subsequently, one square inscribed in every virtual sample plot was used as an image for the dataset, which resulted in 179 images with a resolution of 640 by 640 pixels and a spatial extent of approximately 17.8 by 17.8 m (Table 1). The dataset consisted of 2374 labeled trees, with 551 healthy, 1383 damaged, and 299 dead black pines, 90 other conifers and 51 deciduous trees. The dataset was split into a training set, a development set, and a testing set, using a 4:1:1 ratio for all experiments, while five-fold cross-validation was applied to the combined training and development sets for the final model.

**Table 1.** Details of the labeled dataset.

| Image Size | Number of Images | Number of Labeled Trees | Bands | Spatial Resolution |
|---|---|---|---|---|
| 640 × 640 pixels | 179 | 2374 | RGB | ≈2.8 cm |

*2.2. Algorithm Implementation*

We used the deep learning object detection algorithm YOLOv5 without modifications from the GitHub repository of the same name using the PyTorch framework as of 8 December 2021 [29]. YOLOv5 represents a balanced trade-off between speed and accuracy compared to other commonly used algorithms [30,31]. This algorithm included data augmentation and hyperparameters suited for fine-tuning by default, as well as the capability to ensemble multiple models. The option to automatically generate anchors was disabled for all models trained in this study.

For an easy decision about which model performs better, we chose a single number evaluation metric. On the contrary, having multiple evaluation metrics can make it difficult to decide on one model over the other and may cause rather arbitrary trade-offs between different metrics, such as prioritizing recall over precision or vice versa. The single number evaluation metric used for this paper was the $F_1$ score:

$$F_1 = \frac{2 \times TP}{2 \times TP + FP + FN},$$

(1)

where $TP$ is the number of true positives, $FP$ is the number of false positives, and $FN$ is the number of false negatives. The $F_1$ score can be calculated for single classes, as well as for multiple classes combined, by selecting $TP$, $FP$, and $FN$ accordingly. We chose an intersection over union (IOU) threshold of 0% for correct detections because we considered spatial accuracy to be of low importance for this study. This IOU threshold is different



from the IOU threshold used for non-maximum suppression. It is the minimum overlap required between the predicted bounding box and the ground truth bounding box for the prediction to be considered a correct detection. On the other hand, the IOU threshold used for non-maximum suppression is the threshold used to determine which predicted bounding boxes to keep when multiple bounding boxes are predicted for the same object.

For an optimal non-maximum suppression of the model output, we determined the optimal intersection over union (IOU) threshold and confidence threshold by performing a 16 by 16 grid search, similar to Sirazitdinov et al. [32]. It was performed on the development set, except for the final model, when it was performed on the testing set. The value with the highest $F_1$ score was chosen as the optimal value. Values above an IOU of 0.5 and below a confidence threshold of 0.05 were not tested due to faster processing speeds.

### 2.3. Label Conversion

The chosen object detection algorithm YOLOv5 requires objects to be labeled as bounding boxes. Due to this restriction, the point labels of the dataset exported from ArcGIS Pro in the XLSX file format had to be converted to bounding boxes in the YOLO labeling format. The conversion from points to bounding boxes could have also been performed in ArcGIS Pro. For simplicity, the bounding boxes of every object should be of equal size and square shape. The ideal edge length of the bounding boxes had to be determined. We tested several bounding box edge lengths from 32 to 320 pixels in 32-pixel increments. The edge lengths of 16 pixels below, and above the best-performing length, were also tested. We generated a square bounding box for every tested edge length using the point label as the center. If the bounding box was partially outside the image, it was cropped accordingly so it stayed within the image. For this experiment, the relatively small model size YOLOv5s with 7.2 million parameters was chosen out of training time considerations, because the model had to be trained eleven times. The best edge length was determined by the highest $F_1$ score on the development set, calculated with an IOU threshold of 0% for correct detections. Bounding boxes with this edge length were used for all subsequent models and testing in our study.

### 2.4. YOLOv5 Model Size

We determined the optimal YOLOv5 model size by using the dataset that had the converted labels from the previous experiment, before training the final model. YOLOv5 offers multiple model sizes, which differentiate each other by the model size. To find the best-performing model size, the models YOLOv5n, YOLOv5s, YOLOv5m, YOLOv5l, and YOLOv5x were tested with 1.9, 7.2, 21.2, 46.5, and 86.7 million parameters, respectively [29]. We tested the different model sizes not only on the development set, but also on the training set to detect any potential overfitting behavior. The $F_1$ score calculated with an IOU threshold of 0% for correct detections was used as the evaluation metric.

### 2.5. Final Model Evaluation

We trained and evaluated the final model by utilizing the dataset that included the converted labels and the optimal YOLOv5s model size that we had determined previously. K-fold cross-validation was used for training the final model to improve the performance by using the development set for training as well, while leaving the testing set for final cross-validation. It was important to keep the testing set separate from k-fold cross validation and not use it for either validation during training or training the model to avoid accidental overfitting by the final model. In k-fold cross validation, the data are divided into $k$ equal-sized subsets, or folds. One model was trained on $k - 1$ folds and tested on the remaining fold. This process was repeated $k$ times, with each fold being used as the development set once. We chose five-fold cross-validation because $k = 5$ is the most used value in the literature [33]. We created five YOLOv5 models using this method. They were combined into a single final model utilizing the in-built ensembling feature of YOLOv5. Ensembling

describes the general method of combining multiple models into one final model to obtain a better generalization performance [34].

The final model was evaluated in several ways on the testing set. At first, a confusion matrix was calculated using the same 0% threshold for correct detections that was also used for the $F_1$ score. A confusion matrix enables insight into wrong classifications and might indicate possible measures for improvement.

Afterwards, the $F_1$ scores were calculated for every class, each along their respective 95% confidence intervals. $F_1$ scores as single values are point estimates. Point estimates, sometimes combined with p-values, might suggest meaningful results, although the results might not be very specific. This is opposed to confidence intervals, which can represent the meaning of the result more realistically compared to point estimates [35,36]. Another alternative might be using a point estimate with a standard error [35], which is not viable for the $F_1$ score because of its asymmetric distribution. For this reason, we calculated confidence intervals in addition to the point estimates. The distribution of the $F_1$ score can be sampled using:

$$F_1 = \frac{U}{U+V} \quad \text{with} \begin{cases} U \sim \Gamma(TP + \lambda, 2h) \\ V \sim \Gamma(FP + FN + 2\lambda, h), \end{cases} \tag{2}$$

where $\lambda = 0.5$ and $h = 1$ [37]. A confidence interval with a confidence level of $1 - \alpha$ can be estimated by sampling the distribution from Equation (2) and choosing the $\frac{\alpha}{2}$th percentile for the lower bound and the $(1 - \frac{\alpha}{2})$th percentile for the upper bound, where $\alpha$ is the significance level [38]. A significance level of 5% and thus a 95% confidence interval was chosen for conformity, as those values are the most used values in the literature [39]. We used one million replicates for sampling the distributions.

### 2.6. Extensive Labeling Using the Final Model

The final model ensemble was used for the extensive labeling of one orthomosaic. Due to project constraints, we did not use the final model to label the whole study area. For this, a stride equal to the edge length of the optimal bounding boxes was used. This resulted in overlapping detections. For reducing multiple detections of individual trees, we used the same threshold values determined for the final model with which to perform non-maximum suppression of all combined detections of the orthomosaic. The centers of the resulting bounding boxes were converted back to points, maintaining the class. The resulting label type is analogous to the original dataset. As the deep learning model was trained to identify the tree location and enclose it with identically sized bounding boxes, the predicted centers of these boxes should correspond to the location of the trees. We would not recommend relying on the predicted bounding box size, since the training dataset did not include any information on tree size, and this information is unlikely to be present in the predicted bounding boxes. As a final processing step, all trees outside the region of interest were removed.

## 3. Results and Discussion

### 3.1. Label Conversion

The optimal bounding box size for detecting trees was determined to be 112 pixels (Figure 3a), covering about half of the tree canopy of larger trees and completely covering small trees (Figure 4b). This aligns with other literature such as Yarak et al. [24], who determined that optimal bounding boxes have to cover at least half of the tree canopy. Bounding boxes of this size enabled an effective non-maximum suppression by reducing multiple detections of the same large tree while still being able to differentiate two smaller trees close to each other. Larger bounding boxes performed reasonably well, even if the edge length was equal to half of the image width. This can likely be attributed to the grid search choosing higher IOU threshold values for larger bounding box edge lengths, and thus compensating for the otherwise obvious drop in accuracy to a certain degree. Only

too-small bounding boxes had a significant impact on performance. This might be because small bounding boxes rarely overlap, and thus often solely the confidence threshold can be used for non-maximum suppression.

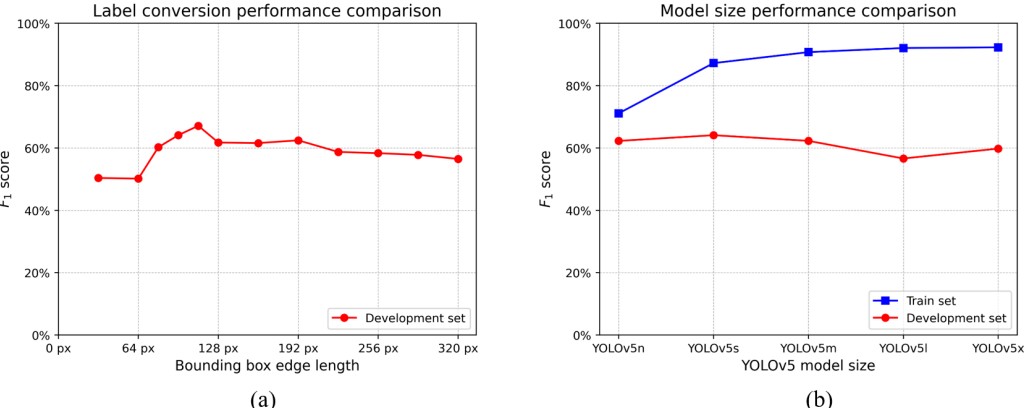

(a)  (b)

**Figure 3.** Results of the experiments. (**a**) $F_1$ scores of YOLOv5s across different bounding box edge lengths in pixels. (**b**) $F_1$ scores on the training set and the development set across multiple YOLOv5 model sizes.

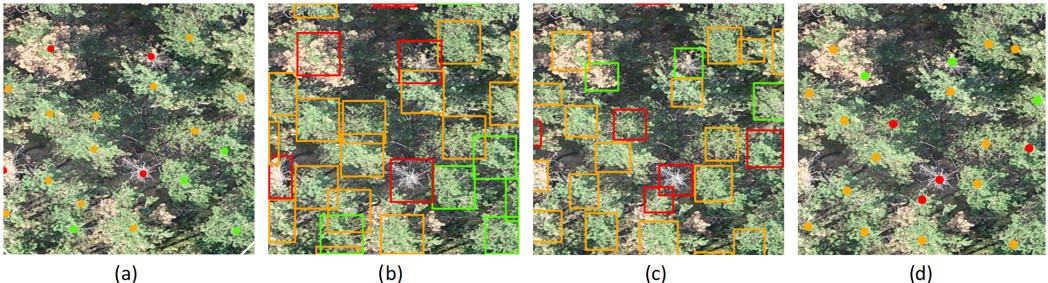

(a)  (b)  (c)  (d)

**Figure 4.** Different labels on the same image with healthy black pines (green), damaged black pines (orange), and dead black pines (red). The image coincidentally contains no deciduous trees or other conifers. (**a**) Point labels from a previous investigation; (**b**) bounding boxes converted from point labels in (**a**) using the optimal edge length; (**c**) output bounding boxes from the final model; (**d**) point labels converted from output bounding boxes in (**c**).

### 3.2. YOLOv5 Model Size

The optimal length of the bounding box edge was used to determine the best-performing YOLOv5 model size. The $F_1$ score on the development set was highest for YOLOv5s, while the performance on the training set increased with model size (Figure 3b). Using a model that is too small may lead to poor performance on both the training and development sets, while a model that is too large may show overfitting behavior. Overfitting denotes the phenomenon of a model optimizing too much to the specific training set and generalizing too little beyond, which also results in a worse performance on the development set. However, it is important to note that the optimal bounding box size was determined with YOLOv5s, which may limit the generalizability of our results to other model sizes. This may have resulted in the dataset giving YOLOv5s an advantage. Using a smaller model such as the optimal YOLOv5s instead of a larger model (e.g., YOLOv5x) also reduces training and inference time significantly, and cuts down the memory requirements. YOLOv5s has 7.2 million parameters, while YOLOv5x, as the largest tested model size, has 86.7 million parameters [29]. This coincidentally makes the following five-fold cross-validation easier to implement.

### 3.3. Final Model Evaluation

Using the optimal bounding box size and the YOLOv5s model, we trained five models using five-fold cross-validation, and these models were ensembled to create the final model using the ensembling built into YOLOv5. The confusion matrix of the ensembled models on the testing set (Figure 5) indicates that our algorithm performs best in detecting damaged black pines, followed by dead black pines, and healthy black pines. Other tree species were not detected correctly, likely due to the low object count in the dataset. Furthermore, healthy black pines were often misclassified as damaged black pines, indicating an unclear distinction in the labeling instructions.

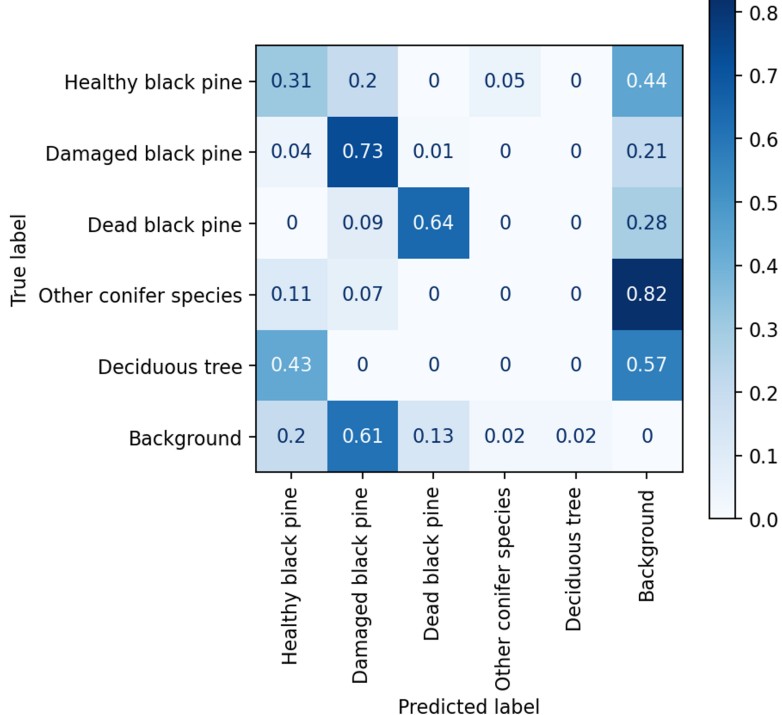

**Figure 5.** Confusion matrix of the final model.

The $F_1$ score point estimates and confidence intervals of our algorithm across different categories are listed in Table 2. The most appropriate literature with which to compare these values of our study are studies covering pine wilt disease on UAV-based imagery, as this disease is the tree disease covered most frequently by studies about monitoring tree disease using UAVs [14]. This disease also covers species within the same genus, *Pinus*, as in our study, and is thus an appropriate comparison. However, one restriction for comparison is that not all articles fitting this description use the $F_1$ score as a metric [40,41].

**Table 2.** $F_1$ scores of the final model across all classes.

| Category | Point Estimate | 95% Confidence Interval |
| --- | --- | --- |
| Healthy black pine | 43% | 32–54% |
| Damaged black pine | 72% | 67–77% |
| Dead black pine | 71% | 60–79% |
| Other conifer species | 0% | 0–20% |
| Deciduous tree | 0% | 1–53% |

Sun et al. [42] proposed a method for RGB imagery that performs tree crown segmentation first and then classifies those segments into the classes tree crown, tree crown discolored by pine wilt disease, and forest gap. The proposed method achieved an $F_1$ score

of 65.8% regarding trees discolored by pine wilt disease (Table 3). Sun et al. [43] compared multiple deep learning object detection algorithms for detecting diseased trees infected with pine wilt disease on RGB images. The compared algorithms were Faster-RCNN, YOLOv4, SSD, YOLOv5, and a proposed algorithm consisting of YOLOv4 with a custom backbone made up of MobileNetv2 with an added CBAM attention and Inceptionv2 structure. The best-performing algorithm was the one they proposed, with an $F_1$ score of 95.6%. The labeled dataset used for this study consisted of 116,012 images, which is hundreds of times larger than the dataset used for our study and might be the main reason for this relatively high $F_1$ score. Only trees infected with pine wilt disease were labeled. Li et al. [44] proposed an algorithm for semantic segmentation on multispectral imagery, based on deep one-class classification, which achieved an $F_1$ score above 90%. The model only detected trees with middle to late-stage pine wilt disease, which were trees with yellow to reddish-brown needles. Xia et al. [45] compared multiple deep learning algorithms for semantic segmentation on RGB images. The best-performing algorithm DeepLLab3+ achieved an $F_1$ score of 82.5%. The segmented labels used for this study covered only trees infected with pine wilt disease.

**Table 3.** Comparison of pine tree vitality monitoring studies utilizing the $F_1$ score as an evaluation metric.

| Study | Algorithm | Task | Spectral Bands | $F_1$ Score for Damaged Pines |
|---|---|---|---|---|
| Sun et al. [42] | Custom algorithm | Segmentation and classification | RGB | 65.8% |
| Sun et al. [43] | Custom YOLOv4 | Object detection | RGB | 95.6% |
| Li et al. [44] | Custom algorithm | Semantic segmentation | Multispectral | >90% |
| Xia et al. [45] | DeepLLab3+ | Semantic segmentation | RGB | 82.5% |
| Our study | YOLOv5 | Object detection | RGB | 67–77% |

The $F_1$ scores of our method fit right within the other literature. From all the studies compared above, each covers mostly one class with definitions most similar to the class of damaged black pine used in our study. The $F_1$ scores achieved in those other studies ranged from 65.8% up to 95.6%, which puts our method with a 95% confidence interval of 67–77% for damaged black pines within this range. Thus, our method achieves competitive accuracy using only point labels converted to bounding boxes. All studies compared above only labeled the tree species under investigation. Similarly, our algorithm might be improved by only labeling black pines, as it performed significantly worse on the classes "other conifers" and "deciduous trees". Increasing the dataset size might also improve our algorithm, similar to Sun et al. [43], who used a significantly larger dataset relative to our study. The effort of labeling a larger dataset is reduced by the efficient labeling of point labels compared to bounding boxes. It is worth noting that both our method and the studies compared above use computationally intensive algorithms, which might limit their applicability in some cases.

The final model was applied to an orthomosaic (Figure 6) using a stride of 112 pixels, which is equal to the optimal edge length determined above. Then, non-maximum suppression was applied to all detections at once (Figure 4c) and the bounding boxes were converted to points (Figure 4d). All points outside the area of interest were deleted. It would be expected that using bounding boxes derived from points loses the information on relative size. This was happening on the one hand for large trees, which were sometimes being detected multiple times and, on the other hand, trees close to each other were sometimes incorrectly detected as one tree. This indicates that a better conversion algorithm for converting points to bounding boxes might improve the results. Algorithms that could be

used for this task could be, for example, seed region growing, watershed delineation [46], layer stacking [47], or PseudoEdgeNet [48].

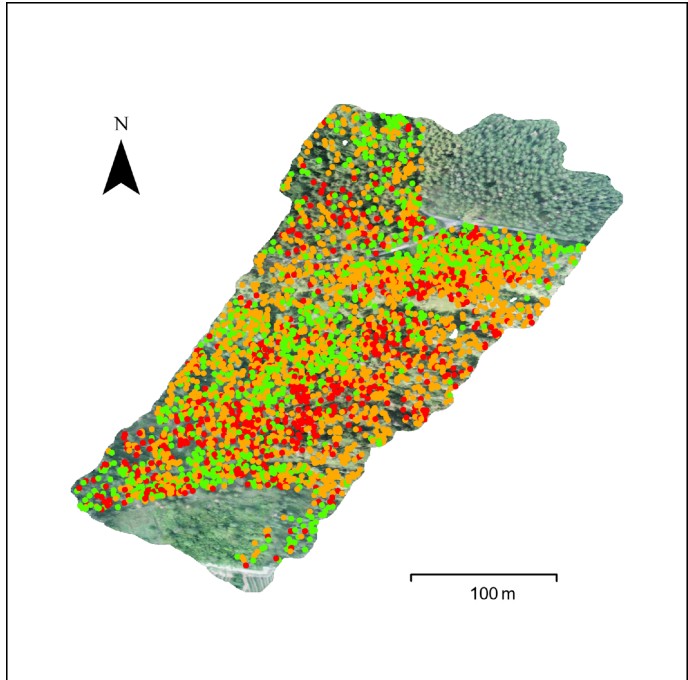

**Figure 6.** The result after post-processing the output of the final model extensively covers the area of interest in one orthomosaic. Each point represents a tree classified as either a healthy black pine (green), a damaged black pine (orange), or a dead black pine (red). The result coincidentally contains no tree classified into another conifer species or a deciduous tree.

## 4. Conclusions

In this study, we proposed a novel method for detecting and classifying damaged trees labeled as point labels using UAV-based orthomosaics. The use of point labels allows for a reduced labeling effort and the opportunity to reuse existing data for training a deep learning algorithm, which is a significant advantage compared to bounding box labels. In the face of globalization and climate change, monitoring forest health is an important tool for pathogen management. Widespread monitoring using remote sensing often requires a dataset that is labeled in a resource-intensive way. We showed that point labels enable more efficient labeling while achieving competitive results. Our proposed method constructs equally-sized square bounding boxes with a point label at the center of each, with the optimal edge length covering at least half of the tree canopy. We found that the YOLOv5s model performed best on the chosen metric and was used for the final algorithm. This study highlights the importance of selecting the optimal bounding box size and model size for achieving the best performance for our approach, as well as the need for clear labeling instructions. Our method was able to label orthomosaics effectively and efficiently. This study provides an efficient and effective way to train a deep learning algorithm for tree vitality monitoring.

Future research could explore using a more sophisticated algorithm for converting point labels to bounding boxes or labeling a larger dataset, which is less resource intensive due to the labels being points. The method proposed in this study enables the native labeling of objects as points or reusing existing data labeled as points combined with unmodified state-of-the-art object detection algorithms, which may enable more effective monitoring of tree vitality. Therefore, this study's significance lies in providing a more efficient and effective way to monitor tree vitality, which can have important implications for the management of forest ecosystems.

**Author Contributions:** Conceptualization, P.H., H.-J.K., S.E., S.R. and J.D.; methodology, P.H.; software, P.H.; formal analysis, P.H.; investigation, P.H.; data curation, P.H.; writing—original draft preparation, P.H.; writing—review and editing, P.H., H.-J.K., S.E., S.R. and J.D.; visualization, P.H.; supervision, S.R.; project administration, H.-J.K. All authors have read and agreed to the published version of the manuscript.

**Funding:** This research received no external funding.

**Data Availability Statement:** The data used for this study are available from the corresponding author, P.H., upon reasonable request.

**Acknowledgments:** The authors thank the Competence Centre for Digital Agriculture of the Weihenstephan-Triesdorf University of Applied Sciences for providing access to a workstation for executing the computationally intensive training.

**Conflicts of Interest:** The authors declare no conflict of interest.

## Abbreviations

The following abbreviations are used in this manuscript:

| | |
|---|---|
| ArcGIS | arc geographic information system |
| IOU | intersection over union |
| RGB | red, green, blue |
| UAV | unmanned aerial vehicle |
| YOLOv5 | you only look once version 5 |

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
