# Peer review of "Application of YOLOv5 for Point Label Based Object Detection of Black Pine Trees with Vitality Losses in UAV Data"

_remotesensing, doi:10.3390/rs15081964_

Round 1

Reviewer 1 Report

find the attached file.

Author Response

Thank you for your valuable feedback, we greatly appreciate it! I have following responses to your comments:

General questions:

Reviewer 1: Why it is important to detect black pine trees.?
Author: I added following sentences: "Black pines were chosen as the research subject because they were believed to be relevant to the regional changing climate in Germany. However, in the largest black pine forest in Germany, many of the trees showed signs of vitality loss."

Reviewer 1: Yolo method is used by many researchers why did the authors choose the yolo family method for detection purposes?
Author: It is as simple as this excerpt of the (revised) manuscript: "[...]as YOLOv5 represents a balanced tradeoff between speed and accuracy compared to other commonly used algorithms"

Reviewer 1: The author should mind their language to improve it for other understanding and reading
Author: I have initiated an extensive english review that gets executed by an experienced colleague of mine.

Introduction:

Reviewer 1:  I have requested for your author to carefully highlight the main contribution to this paper at the end of the introduction, like for example base on this problem they have solved this problem and the main contribution are below
Author: I added this to the end of the Introduction.

Literature:

Reviewer 1: 5 relevant related state-of-the-art techniques
https://doi.org/10.1109/TNSE.2021.3057915 I cited this, as it contains the information that UAVs are low cost
https://doi.org/10.1109/MNET.011.2000154 was not relevant to my manuscript, it covers mostly privacy and big data.
https://doi.org/10.1007/s10489-021-03121-8 was not relevant to my manuscript, it covers mostly crowded human detection.
https://doi.org/10.1109/TGRS.2020.3023135 was not relevant to my manuscript, it covers mostly rural regions and BFTIs
https://doi.org/10.3389/fmars.2022.1086140 I cited this, as it contains the information, that YOLOv5 is fast while being quite accurate.
Author: I really tried to find a fitting place to cite the literature, but only two seemed relevant.

Materials and Methods:

Reviewer 1: If it is possible, please present the dataset detail in the table to easily understand by the reader
Author: I added a table containing the important dataset detail.

Reviewer 1: I am wondering which software they used for labeling and format of the annotation please carefully mentioned in this section if it is possible
Author: We used ArcGIS Pro for labeling and exported the point labels as XLSX. The converted dataset for YOLOv5 is in the YOLO labeling format. I added this information to the revised manuscript.

Reviewer 1: If I am not wrong, in Fig1 if it is the same area how the white place can appear? the UAE can take a picture of the whole area at the same time. The author should have justified this issue or made it the correct figure.
Author: I added this information to the description of the figures. Fig. 1 showed the whole study area consisting of 27 orthomosaics. Fig. 6 showed only one of those 27 orthomosaics.

Reviewer 1: They have used the yolov5 algorithm but they haven’t presented their framework.
Author: YOLOv5 uses PyTorch, i added this information to the revised manuscript.

Reviewer 1: 2.4 section the author is talking about architecture but the just said about the yolov5 different models, the author should present the algorithm presentation in figures for better understanding of reader
Author: I replaced all instances of "architecture" with "model size", as I this might confuse the reader. We only used YOLOv5 models of different sizes.

Reviewer 1: The author has used yolov5 for black pine trees detection but I haven’t found any improvement in the method, the author need should describe in details this section more
logically
Author: I did not modify YOLOv5, but (1) used point labels instead of the usual bounding boxes and (2) applied it to the monitoring use case of black pine vitality assessment. Point labels are the ideal form of labels for this use case coming from first principles, as the information of size contained in bounding boxes is not relevant for monitoring and causes unnecessary overhead during labeling. This was the reason why we experimented with point labels.

Results:

Reviewer 1: There is no comparison between the other methods. It is suggested The author should compare these results with other techniques
Author: I did compare our F1 score, as well as other results to other techniques of different studies, so I assume you meant that I should compare the cited studies amongst each other. I now added a table comparing all studies, including our own.

Discussion:

Reviewer 1: Where is this section? For a scientific paper, it is important to have this section. The author has arranged this section as per the journal template.
Author: We had an internal debate about whether to separate the results and discussion section or combine them. We decided to combine the results and discussion into one chapter, as it led to a better flow of the paper. Some other papers in the forests journal also use this approach, e.g.: https://doi.org/10.3390/f14030451

Conclusion:

Reviewer 1: The conclusion is written week, the author should write first the significance of this study in the first two lines and this describe the problem and then discuss their results Based on these results they should convey some future recommendation.
Author: I changed the whole conclusion to fit your description.

Reviewer 2 Report

Using YOLOv5 and unmanned aerial vehicle (UA V)-based imagery, the authors presents an efficient method for monitoring forest health at the single tree level, 17 using point labels on UA V-based imagery in combination with a deep learning object detection 18 algorithm. The research results can provide information and support for monitoring tree diseases in forests. The article is logically clear, beautifully illustrated, well-expressed, and innovative. However, the following problems exist:

1.       The abbreviations of terms should be given when it first appears, e.g. YOLOv5 in Abstract.

2.       I suggest that you optimize the color matching of the chart to make the representation of the chart clearer and more intuitive. The format of the chart and tables should also be improved.

3.       The authors should further refine their findings and innovations in the discussion section and summarize them based on citations of previous studies.

4.       The manuscript requires English polishing. You need to conduct a thorough edit, addressing technical errors of grammar and wording as well as general readability and style.

Author Response

Thank you for your valuable feedback, we greatly appreciate it! I have following responses to your comments:

Reviewer 2: The abbreviations of terms should be given when it first appears, e.g. YOLOv5 in Abstract.
Author: I now spelled out the full term of YOLOv5 in the abstract, as well as in the introduction chapter. All other abbreviations should be spelled out when they first appear. I used the "acronym" package from latex to generate this automatically.

Reviewer 2: I suggest that you optimize the color matching of the chart to make the representation of the chart clearer and more intuitive. The format of the chart and tables should also be improved.
Author: I changed the color matching and formatting of figure 2 to be more in line with other literature and more intuitive. I added a border outline to the figures containing maps, similar to https://doi.org/10.3390/rs14235936 . Figure 4 already uses the widespread matplotlib for visualization and figure 3 should be clear as it is. I made the table narrower, so it does not look empty anymore. The table formatting itself is the default formatting of the LaTeX MDPI template, for example following paper uses the same table format https://doi.org/10.3390/rs15041157 .

Reviewer 2: The authors should further refine their findings and innovations in the discussion section and summarize them based on citations of previous studies.
Author: I wrote a new subsection "Findings and innovations"

Reviewer 2: The manuscript requires English polishing. You need to conduct a thorough edit, addressing technical errors of grammar and wording as well as general readability and style.
Author: I have initiated an extensive english review that gets executed by an experienced colleague of mine.

Reviewer 3 Report

The authors seek to use an algorithm to detect tree crowns - healthy, damaged, etc. This is an interesting study and shows promise for incorporating AI/ML/Deep learning into forest health assessments. I have a few comments, by section, below.

Abstract:

Line 2:  "...lead to the emergence..."

Comment: Would adding "and spread" be appropriate as you could also use to find range shifts in pathogens, damage, etc.?

Line 3: "...rely on bounding boxes to represent trees." j

Comment: This isn't clear...do the authors mean pixels or non-continuous areas or moving windows for classification/resampling/etc.?

Introduction:

Line 24: "...as it enables to gain..."

Comment: As it enables who? Researchers? Forest resource managers? Seems like a word is missing.

Line 26: Do not think the word Monitoring needs capitalized.

Lines 64-71: Is there a figure to illustrate this? Think that would be helpful - maybe if not here as a summary then in the methods to illustrate how the algorithm works.

Line 79: "...largest black pine forest in Germany..."

Comment: Does the forest have a name?

Materials and Methods

General:

- Did the authors say that they combined the train and dev sets? If so, does that impact the algorithm developing label classifications?

- What does the algorithm do? Some sort of CNN classifying? Also, is it available somewhere - GitHub, purchase, etc.?

Line 86: Same as above - does the forest have a name or a figure showing the location of the 90 ha assessed?

Line 105: Is "dev set" short for developmental dataset for the algorithm? If so, spell out.

Results and Discussion

General:

- Is the algorithm scaling the bounding box as it detects trees/labels? This isn't clear but I may have missed it.

- Can the authors speak to computational/computing/memory requirements? Seems that may be a limiting factor in some cases based on computational resources. 

Line 187: "...edge length was 0.175..."

Comment: Is there a unit?

Lines 189-190: Can the authors provide a range or example of bounding box size(s)?

Lines 223-226: Can the misclassification be overcome? Have other studies had any similar issues? Later studies with more images and greater accuracy are referenced, would this help overcome distinction problems?

Line 233: Might check guidelines for journal but at the beginning of a sentence is it appropriate to use the author name(s) and reference number? Just seeing the number in brackets looks strange.

Author Response

Thank you for your valuable feedback, we greatly appreciate it! I have following responses to your comments:

Abstract:

Line 2:  "...lead to the emergence..."
Comment: Would adding "and spread" be appropriate as you could also use to find range shifts in pathogens, damage, etc.?
Author: I added your suggestion.

Line 3: "...rely on bounding boxes to represent trees." j
Comment: This isn't clear...do the authors mean pixels or non-continuous areas or moving windows for classification/resampling/etc.?
Author: I added the information, that object detection algorithms use bounding boxes. The term of bounding boxes is well known in the research community regarding object detection algorithms.

Introduction:

Line 24: "...as it enables to gain..."
Comment: As it enables who? Researchers? Forest resource managers? Seems like a word is missing.
Author: I paraphrased it for better clarity.

Line 26: Do not think the word Monitoring needs capitalized.
Author: That was an oversight from my side.

Lines 64-71: Is there a figure to illustrate this? Think that would be helpful - maybe if not here as a summary then in the methods to illustrate how the algorithm works.
Author: I created a new figure and added it to the Introduction chapter.

Line 79: "...largest black pine forest in Germany..."
Comment: Does the forest have a name?
Author: No, the public refers to it simply as the black pine forest in the municipalities Leinach and Erlabrunn https://www.mainpost.de/regional/wuerzburg/leinach-deutschlands-groesster-schwarzkiefernwald-in-gefahr-art-10690639

Materials and Methods

General: Did the authors say that they combined the train and dev sets? If so, does that impact the algorithm developing label classifications?
Author: No, all experiments used a fixed 4:1:1 (train, dev, test) split ratio, while the final model uses 5-fold cross validation on the combined train and dev set. Now I added an explaination for better clarity.

General: What does the algorithm do? Some sort of CNN classifying? Also, is it available somewhere - GitHub, purchase, etc.?
Author: It is an object detection algorithm with convolutional layers. I did not mention that it uses convolutional layers, as the inner workings of YOLOv5 are not relevant for my study. My manuscript includes a citation of YOLOv5 https://zenodo.org/record/5563715#.Y_ip3HbMLq4 (This website includes a GitHub link). The models are available on GitHub and we used the GitHub repository for our study.

Line 86: Same as above - does the forest have a name or a figure showing the location of the 90 ha assessed?
Author: I highlighted the study area. The orthomosaics mostly cover the relevant 90 ha. The forest itself extends further than the aerial image suggests.

Line 105: Is "dev set" short for developmental dataset for the algorithm? If so, spell out.
Author: Dev set is a very common name for the dataset used during development, which is why i did not question whether to spell it out, e.g. Stanford uses this notation http://cs230.stanford.edu/blog/split/ . However, you are right, it might not be obvious to the reader. I now have spelled it out.

Results and Discussion

General: Is the algorithm scaling the bounding box as it detects trees/labels? This isn't clear but I may have missed it.
Author: This is a shortcoming of our approach. The original point labels are converted to equally sized bounding boxes, which results in roughly equally sized prediction bounding boxes. Any scaling is only caused by inconsistencies of YOLOv5. This fact is mentioned multiple times throughout the manuscript, I assume that no additional changes are necessary regarding this.

General: Can the authors speak to computational/computing/memory requirements? Seems that may be a limiting factor in some cases based on computational resources. 
Author: I added a sentence about the computational intensity, however this should be given, as mostly all current state of the art algorithms are resource intensive. YOLOv5 in particular is rather hardware agnostic https://github.com/ultralytics/yolov5/issues/7387#issuecomment-1099411744 .

Line 187: "...edge length was 0.175..."
Comment: Is there a unit?
Author: I changed it to pixels, instead of relative width. Converting it to meters would not be ideal, as those would result in values like 2.6762 meters.

Lines 189-190: Can the authors provide a range or example of bounding box size(s)?
Author: I do not completely understand your comment. I specify the bounding box size used (112 pixels) and compare this to other literature, that uses bounding boxes that cover at least half of the tree canopy. I added following passage: "Based on our subjective assessment, the optimal bounding box size determined in the experiment is on average equal to about half of the tree canopy [...]"

Lines 223-226: Can the misclassification be overcome? Have other studies had any similar issues? Later studies with more images and greater accuracy are referenced, would this help overcome distinction problems?
Author: As written in the manuscript, the misclassification indicates unclear labeling instruction regarding the distinction between healthy and damaged black pines. I did not find any other studies closely related to the subject to compare our study to regarding misclassification.  

Line 233: Might check guidelines for journal but at the beginning of a sentence is it appropriate to use the author name(s) and reference number? Just seeing the number in brackets looks strange.
Author: I now changed those citations to \citet{} instead of \cite{}

Reviewer 4 Report

The manuscript “Application of YOLOv5 for point label based object detection of black pine trees with vitality losses in UAV data” proposed a method for detecting diseased black pine trees using point labeled tree as training samples, and tested the optimal bounding box edge length for detecting diseased trees using YOLOv5 models. The proposed method is innovative and effective, will save time for collecting bounding boxes as samples in deep learning process. However, there are problems in the manuscript that need to be addressed and clarification, especially in section 2 materials and methods.

 General comments:

11.  What are the point to convert bounding box to points in the final results? The center of bounding box is not necessarily being the tree location.

22. Description of the methods are not very clear in many places, see specific comments below as example.

33. Section 2 was not well organized. Which step was conducted first, which step was following in YOLOv5 implementation.

44.  “get” and “got” is too colloquial.

Specific comments:

1.        Line 9-10: “Our dataset consisted of 179 images, containing a total of 2374 objects”. Only after read whole manuscript can readers know where these numbers came from. Therefore, this sentence needs clarification.

2.      Line 39-40 “Those algorithms can be divided into either classification, regression, object detection or segmentation.” Grammatical mistake.

3. Line 93-96:  If they are virtual plots, how you determine whether a tree is taller than 2m? how determine the location of a tree? Visual interpolation?

4. There are 5 classes, but in Figure 3 only three, no other conifers and deciduous.

5. Line 101-105: what is “consistent pixel density”? mean pixel size? Are all orthomosaics have same pixel size? What is the spatial resolution?  “with a resolution of 640 by 640 pixels” how big the images? “2374 objects” what the object means here? Labeled trees? The models developed later were performed on each orthomosaic separately or together?

6.        Introduction of YOLOv5 models and their implementation needs clearly description.  Determine optimal bounding box edge length first or determine optimal YOLO v5 architecture first? The final model was used to predict for whole study area?

7.        Line 117 F1 score: this equation is for only one class, how deals with multiple classes because here you have 5 classes.

8.        Set IOU of 0% as threshold seems too loose.

9.        What are the difference among those models (YOLOv5n, YOLOv5s, YOLOv5m, YOLOv5l and YOLOv5x), except for number of parameters?

10.     Line 133 -135 “Several edge lengths were tested with their length being provided in relation to the image edge length. The tested bounding box edge lengths were 0.05 to 0.5 in 0.05 increments. The edge lengths 0.025 below and above the best performing length also were tested.” It is confused, how big the image edge length? How big it is corresponding to the ground? “bounding box edge lengths were 0.05 to 0.5”, what unit they have?

11.     Section 2.5: 5-fold cross validations were performed on 5 models using different bounding box edge length and then selected the best one as final model, is right?

12.     It is better to separate Results and Discussion into different sections.

13.     Line 264: 112 pixels are the bounding box edge length?

Author Response

Thank you for your valuable feedback, we greatly appreciate it! I have following responses to your comments:

General comments:

Reviewer 4: What are the point to convert bounding box to points in the final results? The center of bounding box is not necessarily being the tree location.
Author: I added this information to the subsection "Extensive labeling using the final model". To summarize: The generation of bounding boxes and the conversion back to points is only necessary, as object detection algorithms require bounding boxes. Semantically speaking, the bounding boxes are just point labels of equal spatial extent and should not be confused with actual bounding boxes containing information of size.

Reviewer 4: Description of the methods are not very clear in many places, see specific comments below as example.
Author: I thoroughly went over the Method section and added changes when it seemed appropriate, to make the section clearer.

Reviewer 4: Section 2 was not well organized. Which step was conducted first, which step was following in YOLOv5 implementation.
Author: I now swapped subsection 2.5 with 2.6 to achieve an order analogous to section 3. I added transitional sentences to certain subsections in section 2 to make the order more logical and coherent.

Reviewer 4:  “get” and “got” is too colloquial.
Author: I rephrased all instances of "get" and "got".

Specific comments:

Reviewer 4: Line 9-10: “Our dataset consisted of 179 images, containing a total of 2374 objects”. Only after read whole manuscript can readers know where these numbers came from. Therefore, this sentence needs clarification.
Author: I rephrased it, so it is clear what the images and objects are: "Our dataset used for training and evaluating YOLOv5 models consisted of 179 images, containing a total of 2374 labeled trees."

Reviewer 4: Line 39-40 “Those algorithms can be divided into either classification, regression, object detection or segmentation.” Grammatical mistake.
Author: I removed "either" and changed "divided" to "categorized", which i think is what you meant.

Reviewer 4: Line 93-96:  If they are virtual plots, how you determine whether a tree is taller than 2m? how determine the location of a tree? Visual interpolation?
Author: I used a normalized digital surface model. I now added this information to the Introduction.

Reviewer 4: There are 5 classes, but in Figure 3 only three, no other conifers and deciduous.
Author: This is coincidence. I added that note to the description of the figure.

Reviewer 4: Line 101-105: what is “consistent pixel density”? mean pixel size? Are all orthomosaics have same pixel size? What is the spatial resolution?  “with a resolution of 640 by 640 pixels” how big the images? “2374 objects” what the object means here? Labeled trees? The models developed later were performed on each orthomosaic separately or together?
Author: I rephrased it and added the information, that the resampled pixel density of all orthomosaics is approximately 2.8 centimeters. I changed "objects" to "labeled trees". The extensive labeling using the final model was performed on one orthomosaic.

Reviewer 4: Introduction of YOLOv5 models and their implementation needs clearly description.  Determine optimal bounding box edge length first or determine optimal YOLO v5 architecture first? The final model was used to predict for whole study area?
Author: I added that we used the YOLOv5 repository from GitHub without modifications. We determined the optimal bounding box edge length first (which is why it is the first experiment) and use the generated dataset to determine the optimal YOLOv5 architecture. The final model was used to make predictions one orthomosaic.

Reviewer 4: Line 117 F1 score: this equation is for only one class, how deals with multiple classes because here you have 5 classes.
Author: I added the information, that by choosing TP, FP, and FN accordingly, the F1 score can be calculated for a single class as well as for multiple classes.

Reviewer 4: Set IOU of 0% as threshold seems too loose.
Author: 0 % was chosen, as the spatial accuracy is of low importance for this study. In addition, the resulting optimal bounding boxes only cover about half of the canopies of larger trees. Then 0 % for small-ish bounding boxes seem similar to the 50 % for regularly sized bounding boxes that are a widely used value.

Reviewer 4: What are the difference among those models (YOLOv5n, YOLOv5s, YOLOv5m, YOLOv5l and YOLOv5x), except for number of parameters?
Author: Those are the relevant differences (which is why I only list the number of parameters). Generally speaking, larger models can lead to overfitting, while smaller models can lead to poor performance. If the dataset is large and diverse enough, the bigger the model the better.

Reviewer 4: Line 133 -135 “Several edge lengths were tested with their length being provided in relation to the image edge length. The tested bounding box edge lengths were 0.05 to 0.5 in 0.05 increments. The edge lengths 0.025 below and above the best performing length also were tested.” It is confused, how big the image edge length? How big it is corresponding to the ground? “bounding box edge lengths were 0.05 to 0.5”, what unit they have?
Author: I changed all bounding box edge lengths to pixel values, so this consistent unit can not lead to confusion.

Reviewer 4: Section 2.5: 5-fold cross validations were performed on 5 models using different bounding box edge length and then selected the best one as final model, is right?
Author: No, the 5 models were ensembled using the built in ensembling of YOLOv5. Ensembling describes the process of combining multiple models to get a kind of meta-model in the hope of achieving better generalization. I added an explaination to this section.

Reviewer 4: It is better to separate Results and Discussion into different sections.
Author: We had an internal debate during the writing stage of the manuscript, whether to split the Results and Discussion into different sections. Our consensus was that the combined Results and Discussion section resulted in a better flow of this part of the manuscript.

Reviewer 4: Line 264: 112 pixels are the bounding box edge length?
Author: Yes, I changed all bounding box edge lengths to pixel values, so this consistent unit can not lead to confusion.

Round 2

Reviewer 1 Report

I humbly thank the authors' efforts on the present manuscript. I think the manuscript is significantly improved.  I agree it can be published in remote sensing

Author Response

Thank you for this message! However, please only accept the revision instead of writing a comment. Otherwise, the MDPI Susy website interprets this as a call for a new revision.

Reviewer 4 Report

The revised version is much improved, especially the description of method is clearer. But there are still many problems, and further refinement is needed for a high quality paper. Below is some questions and suggestions. 

 1. The optimal bounding box edge length was determined using YOLOv5s. Is any possible to obtain different optimal edge lengths using different models?

2. YOLOV5s was identified as the best model. Do you ever thought that use of the optimal bounding box edge length determined by YOLOv5s might be the reason? How about if using the optimal length determined by other model?

3. Section 2.5 Final model evaluation: is cross-validation performed during the training process only using training and develop set or using all dataset? Usually, the final model is evaluated using independent testing set, producing confusion matrix, then calculate accuracy measurement metrics, for example, overall accuracy, user’s and producer’s accuracy, Kappa. For deep learning approach, it might use different metrics like F1 score.

4. The model was developed using samples collected from whole study area (Figure 2), thus the model should be applied to all RGB images, producing a map showing identified tree classes. Then analyze the spatial distribution of total infected trees. Of course, for better visualization, you can enlarge a part of it.  

4. Section 3.4 Findings and innovations: Most of this section is repetition of section 3.1-3.3. suggest combining section 3.4 and conclusion together.

5. Some paragraphs in section of result and discussion are too verbose. Suggest re-write them and make them concise and to the point.

6. English writing needs improvement.

 Specific:

1. we call samples used to train and test a model as training set and testing set, not train set.  

2. Line 140-141: “As an example for comparison, the famous PASCAL Visual Object Classes challenge uses an intersection over union (IOU) threshold of 50 % for correct detections [32].” I did not see comparison in this study. If not, this sentence can be deleted. In addition, the following paragraph,”…we determined the optimal IOU…”. As I read, this study used 0% as threshold of correct detection, no optimal IOU was identified.

3. Line 151-153: Converting point labels to square polygons is very simple in ArcGIS, thus you can do it in ArcGIS.

4. Line 166-167: please revise the first sentence to make it clear.

5. Line183: insert into between models and one final model.

6. section 2.5: Based on what dataset the model evaluation was performed? Training/development or testing sets?  Make this section clear.

7. Figure 4. Caption: grammar errors, suggest: Point labels from a previous investigation; (b) bounding boxes converted from point labels in (a) using the optimal edge length; (c) output bounding boxes from the final model; (d) point labels converted from output bounding boxes in (c).

Author Response

We appreciate your valuable feedback and thank you again! I have carefully read your comments and answered them point by point:

General comments:

Reviewer 4: The optimal bounding box edge length was determined using YOLOv5s. Is any possible to obtain different optimal edge lengths using different models?
Author: That is a very good point and might be a potential shorcoming of our approach. I added a passages about this to the discussion section.

Reviewer 4: YOLOV5s was identified as the best model. Do you ever thought that use of the optimal bounding box edge length determined by YOLOv5s might be the reason? How about if using the optimal length determined by other model?
Author: See response above. I added two passages about this to the discussion section. The generalizability of the results might be limited to some extent. However we did not use multiple model sizes for determining the optimal bounding box length, as it would require each model size to be trained 11 (!) times using our methodology. "For this experiment, the relatively small model size YOLOv5s with 7.2 million parameters was chosen out of training time considerations, because the model had to be trained eleven times."

Reviewer 4: Section 2.5 Final model evaluation: is cross-validation performed during the training process only using training and develop set or using all dataset? Usually, the final model is evaluated using independent testing set, producing confusion matrix, then calculate accuracy measurement metrics, for example, overall accuracy, user’s and producer’s accuracy, Kappa. For deep learning approach, it might use different metrics like F1 score.
Author: We kept the test set separate to obtain independent cross-validation results for the F1 score. I added this sentence: "It was important to keep the test set separate from k-fold cross validation and not used for either validation during training or training the model to avoid accidental overfitting to the final model."

Reviewer 4: The model was developed using samples collected from whole study area (Figure 2), thus the model should be applied to all RGB images, producing a map showing identified tree classes. Then analyze the spatial distribution of total infected trees. Of course, for better visualization, you can enlarge a part of it.  
Author: Due to project constraints, we did not apply the model to the whole study area. I added this information to the Results and Discussion section.

Reviewer 4: Section 3.4 Findings and innovations: Most of this section is repetition of section 3.1-3.3. suggest combining section 3.4 and conclusion together.
Author: I followed your advice, deleted section 3.4, and combined relevant information from section 3.4 with the conclusion into a rewritten conclusion.

Reviewer 4: Some paragraphs in section of result and discussion are too verbose. Suggest re-write them and make them concise and to the point.
Author: I rewrote many paragraphs in the Result and Discussion section to reduce verbosity.

Reviewer 4: English writing needs improvement.
Author: An experienced colleague of mine performed english corrections to the newly revised manuscript. Those changes are visible in the manuscript, as they were performed by "JanD".

Specific comments:

Reviewer 4: we call samples used to train and test a model as training set and testing set, not train set.  
Author: I changed all instances of train set to training set and all instances of test set to testing set.

Reviewer 4: Line 140-141: “As an example for comparison, the famous PASCAL Visual Object Classes challenge uses an intersection over union (IOU) threshold of 50 % for correct detections [32].” I did not see comparison in this study. If not, this sentence can be deleted. In addition, the following paragraph,”…we determined the optimal IOU…”. As I read, this study used 0% as threshold of correct detection, no optimal IOU was identified.
Author: I removed the sentence you cited. The IOU threshold of 0 % for correct detections is an IOU threshold different from the IOU threshold used for non-maximum suppression. I added a passage for clarification.

Reviewer 4: Line 151-153: Converting point labels to square polygons is very simple in ArcGIS, thus you can do it in ArcGIS.
Author: I now added the passage "The conversion from points to bounding boxes also could have been performed in ArcGIS Pro", as I did not do it in ArcGIS Pro, but it would have been possible.

Reviewer 4: Line 166-167: please revise the first sentence to make it clear.
Author: I revised it: "We determined the optimal YOLOv5 model size by using the dataset that had the converted labels from the previous experiment, before training the final model."

Reviewer 4: Line183: insert into between models and one final model.
Author: I revised the sentence for better readability: "Five YOLOv5 models were created using this method, and they were combined into a single final model utilizing the in-built ensembling feature of YOLOv5."

Reviewer 4: section 2.5: Based on what dataset the model evaluation was performed? Training/development or testing sets?  Make this section clear.
Author: I made it more clear by this: "The final model was evaluated in several ways [on the testing set]." and that sentence: "It was important to keep the testing set separate from k-fold cross validation and not used for either validation during training or training the model to avoid accidental overfitting by the final model." 

Reviewer 4: Figure 4. Caption: grammar errors, suggest: Point labels from a previous investigation; (b) bounding boxes converted from point labels in (a) using the optimal edge length; (c) output bounding boxes from the final model; (d) point labels converted from output bounding boxes in (c).
Author: I followed your suggestion